# Career Calling, Courage, Flourishing and Satisfaction with Life in Italian University Students

**DOI:** 10.3390/bs13040345

**Published:** 2023-04-20

**Authors:** Anna Parola, Andrea Zammitti, Jenny Marcionetti

**Affiliations:** 1Department of Humanities, University of Naples Federico II, 80133 Naples, Italy; 2Department of Philosophy, Sociology, Education and Applied Psychology, University of Padua, 35139 Padua, Italy; 3Department of Education and Learning, University of Applied Sciences and Arts of Southern Switzerland, 6928 Manno, Switzerland; jenny.marcionetti@supsi.ch

**Keywords:** career calling, courage, well-being, university students

## Abstract

Career calling is defined as a positive resource promoting vocational development and well-being. The present study focuses on the relationships between career calling, courage and two indicators of well-being, i.e., flourishing and satisfaction with life. The sample consisted of 306 Italian university students (118 males and 188 females) ranging from 18 to 30 years of age. A structural equation modeling (SEM) approach with latent variables was adopted. The results showed that courage plays a mediating role between career calling and well-being indicators. In light of these results, suggestions on the practical implications for career interventions to support university students are also addressed.

## 1. Introduction

Career calling is a developmental construct that is especially important for young adults when making career choices. Setting their career goals and making career choices are crucial tasks during the university period. Moreover, career choices seem to inform one’s overall sense of meaningfulness in life.

The challenges that characterize the current society (e.g., globalization, economic crises, pandemics, wars) are considered risk factors for developing young people’s ability to think about their future. This study addresses the role of career calling as a booster of positive outcomes. In countries with stagnant economies such as Italy [1], feeling one’s career as a calling could be a driving force to implement adaptive career-related behaviors that promote well-being. Moreover, many studies support the role of courage among the psychosocial resources that could help people face these challenges. In particular, Ginevra et al. [2] consider the role of courage as an adaptive behavior to cope with career development tasks and changing work, and to promote life satisfaction.

On these premises, this study focuses on the relationship between career calling and well-being and the possible mediating role of courage as a protective factor in the career choice process. While the relationship between career calling and well-being has been addressed [3,4], the role of courage in this relationship is still unexplored, and this study aims to fill this gap.

### 1.1. Theoretical Framework

Building career paths is a major task in adolescence and emerging adulthood [5,6,7]. Today’s working scenario is characterized by numerous challenges and changes that put a strain on people’s abilities and possibilities to project their working future. In a complex scenario, new theories and modes of intervention emerge to help people read their surroundings [8]. According to career construction theory (CCT; [9]), people construct their careers by giving meaning to their experiences. This theory is interested in the more subjective aspect of a career being understood as a construct that gives meaning to past and present experiences and future aspirations. Thus, career construction passes through attributing meaning to the subject’s experiences. The process of developing one’s career path seems to involve different personal resources that could help one cope with the difficulties of the future [10].

As some studies have shown [11,12], the transition from university to the world of work is favored by certain socio-demographic and personal variables. However, additional psychological resources can facilitate this transition; career calling and courage are among these resources.

According to Duffy and colleagues [13], a definition of career calling reflects a multidimensional nature that comprises components related to meaning, prosocial motivation and internal/external summons. Meaning refers to finding personal meaning and purpose in work. Prosocial motivation indicates helping others or contributing to the common good. Finally, internal/external summons refers to feeling compelled to perform a certain type of work. Calling represents a career orientation [14] and may affect early career development [15]. University often coincides with the developmental stage of emerging adulthood, in which the major developmental tasks are related to establishing an identity [16], including making significant career decisions. Emerging adulthood is defined as the life stage between adolescence and adulthood, lasting from the ages of 18 to about 29 [17]. According to the latest theoretical conceptualization of established adulthood provided by Mehta [18], in one’s 20s, a great deal of effort is devoted to obtaining an education and occupational sampling paths before making an occupational commitment around the age of 30.

The presence of a calling is described as a positive resource that promotes vocational development [19,20]. A calling may help individuals to activate and anticipate the planning of their careers [21]. In this sense, a calling is linked to future-oriented identity construction because it provides a sense of direction toward one’s purpose in life [22]. According to Hirschi and Herrmann [15], a calling enhances a person’s confidence in mastering challenges at work and finding ways to actualize one’s callings in their work role. Previous studies provided evidence of the presence of a calling among university students [3,23], supporting the role of a calling in the career decision-making process [23].

Looking at one’s career as a calling is linked to the outcomes in career development and well-being. Several studies focused on the relationship between career calling and well-being among university students [3,4,24,25] and adult workers [26]. According to Duffy and Dik [27], the link between career calling and life satisfaction is more pronounced in adults than in university students. Duffy and Dik [27] proposed an interpretation of this assumption by taking into account the evidence that adults are more likely to be in the world of work than university students, and by considering that mediators and moderators might play a role in the relationship between career calling and life satisfaction. For example, Duffy et al. [25] showed how academic satisfaction and life meaning fully explain the relationship between career calling and life satisfaction.

We have chosen flourishing and life satisfaction as the well-being outcomes. While the broadest life satisfaction refers to a global measure of satisfaction, the flourishing concept, in a positive psychology framework, is defined as an optimal state of functioning in which people pursue personal goals and aspirations [28]. The optimal state of functioning involves competence, self-acceptance, meaning and relatedness, optimism, giving and engagement. According to Seligman [28], flourishing enhances the possibility of having a successful career. Concerning university students, Van Zyl and Stander [29] recommend enhancing the role of flourishing in academic careers.

In this process, adaptive behaviors for coping with career development tasks can mediate between career calling and well-being. Among other behaviors, courage could play a critical function. However, defining courage is not an easy task. Indeed, different fields have tried to define this construct, from philosophy [30] to psychology [31,32,33,34] and the field of nursing [35,36,37].

From a psychological perspective, some authors have described courage by linking it to fear. For example, one of the most known definitions of courage is the one put forward by Norton and Weiss [38], who state that courage is “persistence or perseverance despite having fear” (p. 213). Rate, Clarke, Lindsay and Sternberg [39] identified four main dimensions of courage; it is (1) an interactional action (2) characterized by the presence of fear, (3) nobility of purpose and (4) personal risk. Woodard [40] also defined courage as “the ability to act for a meaningful (noble, good, or practical) cause, despite experiencing the associated fear” (p. 174). The conceptualization of courage as the propensity to act despite the presence of fear plays an important role in today’s social context, which can help people face challenges [41], act to achieve a positive outcome [42] and protect what is deemed important, including one’s career [43]. This could imply that courage is a positive mediation between career-related dimensions and personal well-being.

The mediating role of courage and the link between courage and well-being is widely recognized in the literature. For example, in Bockorny’s study [44], courage correlates positively with life satisfaction; Gustems and Calderon’s [45] and Toner and colleagues’ [46] studies emphasized a correlation between courage and psychological well-being. In addition, Magnano et al. [43,47] found that courage mediates the relationships between certain dimensions related to career planning or meaningful work and personal well-being; Ginevra et al. [48] have also shown that courage mediates the relationship between career adaptability and life satisfaction.

### 1.2. Current Study

Based on the literature review above, career calling can be regarded as a fundamental psychological construct related to life satisfaction. Furthermore, courage has been found to play a role in coping with career development tasks in university students by promoting well-being. Therefore, this study aims to explore the role of courage in the relationship between career calling and two measures of well-being, life satisfaction and flourishing. We assume that courage motivates people to protect what they consider to be valuable. Therefore, if individuals experience their careers as a vocation, this encourages them to protect their careers, which may impact their well-being. Specifically, we expected the following:

**Hypothesis** **1 (H1):**
*The relationship between career calling and flourishing is mediated by courage.*


**Hypothesis** **2 (H2):**
*The relationship between career calling and satisfaction with life is mediated by courage.*


## 2. Materials and Methods

### 2.1. Participants and Procedure of Data Collection

The sample included 306 participants (118 males and 188 females) from the ages of 18 to 30 (M = 22.6; SD = 1.9). The participants were distributed as follows: 16.7% north; 34.5% center; 43.8% south; 5% islands.

Data were collected from Italian university students using the snowball sampling method. The participants were recruited from the general population through advertisements on social media. According to the inclusion criteria, the participants had to (A) be between 18 and 30 years of age, (B) be a current university student, (C) be a native Italian speaker and (D) provide informed consent.

The participants were informed about the anonymity and confidentiality of the survey and the possibility of leaving at any time.

### 2.2. Measures

Flourishing

The Italian version of the Flourishing Scale [49] was used to assess flourishing. The scale consists of 8 items (e.g., “I lead a purposeful and meaningful life”) rated on a 7-point Likert scale ranging from 1 (strongly disagree) to 7 (strongly agree). Cronbach’s alpha in the study sample was 0.90.

Life satisfaction

The Italian version of the Satisfaction with Life Scale [50] was used to assess satisfaction with one’s life. The scale consists of 5 items (e.g., “In most ways my life is close to my ideal”) rated on a 7-point Likert scale ranging from 1 (strongly disagree) to 7 (strongly agree). Cronbach’s alpha in the study sample was 0.93.

Courage

The Italian version of the Reduced Courage Measure [2] is based on the definition of courage by Norton and Weiss [38] and was used to assess courage. The scale consists of 6 items (e.g., “If there is an important reason to face something that scares me, I will face it”) rated on a 7-point Likert scale ranging from 1 (never) to 7 (always). Cronbach’s alpha in the study sample was 0.85.

Career calling

The Unified Multidimensional Calling Scale (UMCS; [51]) was used to assess the presence of career calling. The UMCS consists of 22 items divided into 7 dimensions including Passion (4 items, e.g., “I am passionate about what I am studying”), Sacrifice (3 items, e.g., “I can give up many things to keep studying this subject”), Transcendent Summons (3 items, e.g., “I am pursuing this line of study because I believe I have been called to do so”), Prosocial Orientation (3 items, e.g., “I always consider how beneficial my work will be to others”), Pervasiveness (3 items, e.g., “My current line of study is always on my mind), Purposefulness (3 items, e.g., “My academic and professional career is important to give meaning to my life”) and Identity (3 items, e.g., “What I study is part of who I am”). The items were rated on a 7-point Likert scale ranging from 1 (strongly disagree) to 7 (strongly agree). Cronbach’s alpha in the study sample was 0.95 for the whole dimension.

### 2.3. Data Analysis

The Pearson correlation coefficient (r) was computed to evaluate the relationships between variables.

A structural equation modeling (SEM) approach with latent variables was adopted. In the first step, a partially disaggregated parcel approach was used. Item parcels were used as indicators of latent variables [52,53], and at least 3 item parcels were created for each latent variable [53,54]. To create parcels of the three unidimensional scales, courage, flourishing and satisfaction with life, the item-to-construct balance strategy [53] performed by inspecting factor ladings resulting from each measurement model was used. To create parcels of the hierarchical second-order structure of the career calling measure, the domain representative strategy [53,55] performed by aggregating together items of each dimension was used.

In the second step, a mediation model with latent variables was set by using a two-step approach. In step 1, a predictor-only model was specified; the career calling (X) was regressed on flourishing (Y1) and satisfaction with life (Y2). In step 2, the full mediation model was specified; the career calling (X) was regressed on flourishing (Y1) and satisfaction with life (Y2) through courage (M) (Figure 1). The MLR estimator was used.

The model was assessed using the chi-square statistics (χ^2^), the root mean square error of approximation (RMSEA), the comparative fit index (CFI), and the standardized root mean residual (SRMR). The cut-off criteria used to evaluate the goodness of fit includes statistically non-significance of the χ^2^, an RMSEA and SRMR lower than 0.08, and a CFI higher than 0.90 [53,56].

## 3. Results

The correlation analysis showed small-to-large associations between the variables involved in the mediation model (Table 1).

The hypothesized model (Figure 2) provided a good fit to the data χ^2^ (94) = 260.019; *p* < 0.001; RMSEA = 0.076; 90% CI: [0.065, 0.087]; CFI = 0.935, SRMR = 0.051.

All the item parcels showed a factor loading higher than 0.66. The analysis showed that career calling predicts courage, flourishing and life satisfaction. Moreover, courage predicts both flourishing and life satisfaction.

The indirect effect “career calling → courage → flourishing” was significant (*β* = 0.170, SE = 0.052, *p* < 0.001) as well as the indirect effect “career calling → courage → life satisfaction” (*β* = 0.146, SE = 0.053, *p* < 0.001).

The total model effect was significant both for Y1 (*β* = 0.786, SE = 0.058, *p* < 0.001) and Y2 (*β* = 0.547, SE = 0.040, *p* < 0.001).

## 4. Discussion

This study aims to understand the relationship between career calling and two well-being outcomes, life satisfaction and flourishing, and to see whether courage could mediate the relationship between career calling and the two well-being outcomes. The model tested suggests the meaningfulness of a partial mediation model from career calling to life satisfaction through courage, and from career calling to flourishing through courage.

Career calling helps people to become active in their career planning [21]. On the other hand, courage—defined as the tendency to act despite fear [38]—is a resource that many researchers have recently anchored in the field of career development. In a society characterized by multiple challenges, courage helps people to protect what they consider to be important, including their career [43]; in the post-pandemic society, courage helps people to overcome fears that can block them in their career planning [57]. Furthermore, courage helps people to build positive career paths and cope with various career and life decisions [58]. In this sense, considering one’s career as a calling can activate courageous, protective behavior toward one’s professional projects. This would help people to cope with the challenges they face in planning their futures. Moreover, with its mediating role, courage can be translated into behaviors that contribute to a perceived good level of personal well-being. Indeed, other research also shows that courage can activate adaptive behaviors that promote life satisfaction and flourishing [48,59]. Similarly, feeling one’s career as a calling would support one in perceiving a good life, as confirmed by other studies [2,3,24,25].

A critical reading of these findings can be made in light of those approaches that emphasize the importance of the subjective evaluation of professional success and advocate enhancing personal resources to help design the future. From a developmental point of view, improving psychological resources to overcome career challenges can be supported by activities guided by career construction theory [9]. Career construction theory is the theoretical basis for life design that is a narrative-based activity aimed at supporting individuals to create a personal meaning about life plans and career decisions. The life design approach argues that in today’s complexity, people are free to manage their career paths more than they were in the past, and need to make sense of the challenges through their resources [8]. The interventions focus on enhancing the psychological resources that an individual need to create careers in today’s society. Enhancing the skills related to self-reflection and engaging in basing plans on identity is the mission of life design [60].

In the evaluation of professional success, there has been a focus on objective evaluations (e.g., status or income) for a long time. However, there is also a subjective evaluation of professional success, characterized by the satisfaction that individuals derive from their work or the contribution their work makes in giving meaning to their lives. Since career calling is also subjective passion and satisfaction, Hall and Chandler [20] suggest that career calling can be a form of subjective career success. Experiencing one’s job as a calling can contribute to experiencing well-being and subjective success, even in the absence of objective success. Finally, courage plays a protective role in dealing with career-related tasks [61]; thus, it can reduce negative emotions and help people to activate protective behavior toward their careers, allowing them to experience personal well-being.

The limitations of this study must be acknowledged. First, the cross-sectional nature does not permit the inference of reliable causal relationships between the study variables. Longitudinal studies are needed to test the relations found. Second, all the measures used were self-reported, and well-known reporting biases, such as social desirability, may influence the data. In the future, researchers could use a mixed approach to data collection (quantitative and qualitative) to allow for a more in-depth study of career calling. Third, the sampling presents a high prevalence of females, and a multi-group analysis comparing the model across males and females was not computed due to the unbalanced sample. Further studies should investigate if there are differences between males and females in the relationships between the study variables.

It is also important to note that this study is conducted in the Italian context. Therefore, the results should also be read considering the context and cultural aspects. Some studies revealed that cultural aspects could affect the exact definition of career calling, for example, between Eastern and Western cultures, suggesting that the sense of calling is probably not a universal concept [62]. This evidence reinforces the need for cross-cultural research [14,63].

Despite these limitations, this study contributes theoretically and practically to existing research in the career field. From our point of view, the assessment of career calling can be used for career interventions. Work that is felt as a ‘vocation’ leads to positive effects in terms of well-being that cannot be overlooked when working with adolescents and young adults in planning possible career paths. As found in our study, if individuals experience their careers as a vocation, this encourages them to protect their careers, which has a positive impact on their well-being. Furthermore, experts working in the field of career counseling should know that career calling and courage are not stable dimensions of personality, but can change. This means that career development training, following a life design perspective, can be designed in such a way as to stimulate courage or the perception of work aspects that make one’s job satisfying.

## Figures and Tables

**Figure 1 behavsci-13-00345-f001:**
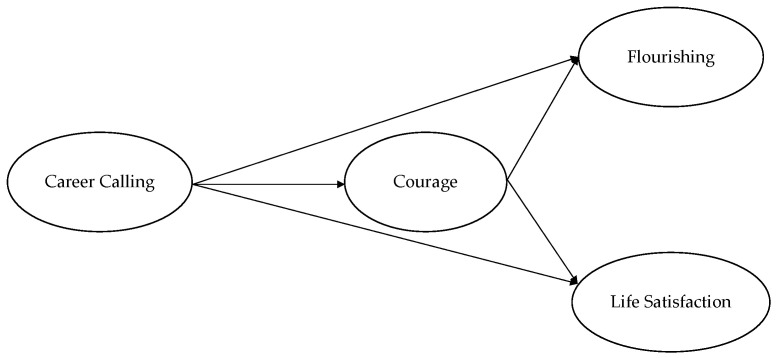
Hypothesized model.

**Figure 2 behavsci-13-00345-f002:**
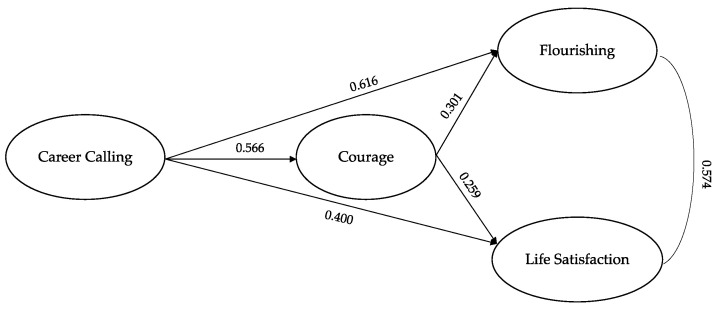
Mediation model.

**Table 1 behavsci-13-00345-t001:** Means, standard deviations and correlations among study variables.

		M	SD	1	2	3	4	5	6	7	8	9	10
1	FS	5.82	0.89	-									
2	LS	4.96	1.32	0.688	-								
3	CS	5.56	0,96	0.543	0.420	-							
4	CALL_PASS	5.63	1.17	0.674	0.555	0.342	-						
5	CALL_SAC	5.57	1.26	0.615	0.500	0.440	0.786	-					
6	CALL_TrS	4.95	1.68	0.506	0.396	0.352	0.557	0.531	-				
7	CALL_PRO	5.83	1.13	0.532	0.347	0.322	0.492	0.497	0.584	-			
8	CALL_PER	4.87	1.46	0.458	0.318	0.307	0.581	0.584	0.479	0.513	-		
9	CALL_PUR	5.67	1.13	0.444	0.216	0.377	0.499	0.556	0.349	0.458	0.573	-	
10	CALL_IDE	5.67	1.12	0.597	0.349	0.464	0.587	0.595	0.554	0.518	0.615	0.709	-
11	CALLING	5.46	1.01	0.691	0.489	0.470	0.815	0.823	0.766	0.735	0.797	0.734	0.822

Note. M = means; SD = standard deviation; FS = flourishing; LS = life satisfaction; CS = courage; CALL = calling with; PASS = passion; SAC = sacrifice; TrS = transcendent summons; PRO = prosocial orientation; PUR = purposefulness; and IDE = identity. All correlations are statistically significant with *p* < 0.001.

## Data Availability

The data presented in this study are available upon reasonable request from the corresponding author. The data are not publicly available due to privacy reasons.

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
