# Peer review of "Career Calling, Courage, Flourishing and Satisfaction with Life in Italian University Students"

_behavsci, 2023, doi:10.3390/bs13040345_

Round 1
Reviewer 1 Report
The article is written at a high fundamental level.
The theoretical framework includes a logical justification of the selected sources to confirm its hypotheses. Hypotheses and methods are fully justified. The main methods of analysis are professionally described, and a reliable model is built that fully substantiates the proposed hypotheses. The authors are fully aware of the debatable nature of the proposed research, which they logically set out in the Discussion section and drew reasoned conclusions
Author Response
We would like to thank the reviewer for reading our article.
We are glad for all the positive feedback!
Please note that, as requested, one of our native English-speaking colleagues has checked the paper.
Reviewer 2 Report
I would first like to thank the authors for the opportunity to read their work, Career calling, courage, flourishing and satisfaction with life in Italian university students. I found the manuscript to be very well written and a valuable contribution to our understanding of career decision-making. Below are a few clarifying questions that would help the reader further understand the goals of the manuscript.
Comment 1: Briefly define key variables in the introduction (i.e., career calling, courage)
Comment 2: The literature review indicates the focus of the study is on emerging adults. Yet, the study’s sample ranges from 19-30 years of age. When is an individual believed to be in the “emerging adult” stage?
Comment 3: The paper describes several dimensions to courage. Does the Reduced Courage Measure capture those dimensions? If not, what is the rational for using this specific scale?
Comment 4: The steps of analysis are clearly described. Thank you for the transparency of your method.
Comment 5: How might the findings of the current study be impacted by cultural differences? Might there be something specific about this population that related to the variables differently than other populations?
Author Response
We thank the reviewer for their suggestions and for appreciating our paper. Below is our point-by-point response.
Comment 1: Briefly define key variables in the introduction (i.e., career calling, courage)
Thank you for this comment. We have specified our key variables in the introduction (in yellow in the text).
Comment 2: The literature review indicates the focus of the study is on emerging adults. Yet, the study’s sample ranges from 19-30 years of age. When is an individual believed to be in the “emerging adult” stage?
Thank you for this comment. We have included the most recent definition of emerging adulthood provided by Arnett and colleagues (2014) and added the age range (in yellow in the text).
Comment 3: The paper describes several dimensions to courage. Does the Reduced Courage Measure capture those dimensions? If not, what is the rational for using this specific scale?
In the description of the scale we have added that the scale is based on the definition of Norton & Weiss (2009), which was reported in the introduction.
Comment 4: The steps of analysis are clearly described. Thank you for the transparency of your method.
Thank you!
Comment 5: How might the findings of the current study be impacted by cultural differences? Might there be something specific about this population that related to the variables differently than other populations?
Thank you. We have addressed this issue in the discussions
Reviewer 3 Report
Comments on the article “Career calling, courage, flourishing and satisfaction with life in Italian university students” submitted to Behavioral Sciences
Thank you for inviting me to review this interesting manuscript. It is written very well and in a neat way. It provide findings suggesting the meaningfulness of a partial mediation model from career calling to life satisfaction through courage and from career calling to flourishing through courage.
Below you will find my comments and tips on how to improve the manuscript.
1. The title and abstract cover the main aspect of the work. They include adequate names of the examined variables. The abstract briefly and accurately describes the results of the study.
2. The Introduction is written clearly and logically, both concisely and comprehensively. It provides definitions and results of previous research on the analyzed variables. However, there is no clear theoretical justification for testing exactly career calling, courage, flourishing and satisfaction with life in young people. I would recommend the use of one of the psychological theories that would allow the Authors to justify their research. For instance, the Authors may find Erik Erikson's theory of psychosocial development relevant for their study. A review of the findings so far seems insufficient to formulate a comprehensive theoretical background.
3. The method is adequate to the hypotheses put forward by the Authors. The statistical analysis is appropriate to the research.
4. Please, correct Table 1 by using dots instead of commas.
5. The discussion could have been better organized if the Authors had used a guiding psychological theory (as mentioned earlier) to interpret their findings
6. Are there more precise practical implications of the study? Please provide a few.
Author Response
We thank the reviewer for their suggestions. Below is our point-by-point response.
- The title and abstract cover the main aspect of the work. They include adequate names of the examined variables. The abstract briefly and accurately describes the results of the study.
- The Introduction is written clearly and logically, both concisely and comprehensively. It provides definitions and results of previous research on the analyzed variables. However, there is no clear theoretical justification for testing exactly career calling, courage, flourishing and satisfaction with life in young people. I would recommend the use of one of the psychological theories that would allow the Authors to justify their research. For instance, the Authors may find Erik Erikson's theory of psychosocial development relevant for their study. A review of the findings so far seems insufficient to formulate a comprehensive theoretical background.
We would like to thank the review for allowing us to deepen our psychological theoretical framework of reference (Savickas' Career Construction Theory). Thank you, also, for allowing us to anchor our framework even more deeply in psychology by also introducing the importance of Erikson's identity studies that paved the way for the possibility of viewing career choices as a developmental task.
- The method is adequate to the hypotheses put forward by the Authors. The statistical analysis is appropriate to the research.
- Please, correct Table 1 by using dots instead of commas.
Done. Thank you.
- The discussion could have been better organized if the Authors had used a guiding psychological theory (as mentioned earlier) to interpret their findings
We have revised the results in light of the explanations of the theoretical framework. Thank you!
- Are there more precise practical implications of the study? Please provide a few.
Thank you for this suggestion. We have deepened the practical implications in the final part of the manuscript.